

# Exploring the relationship between surface PM$_{2.5}$ and meteorology in Northern India

Jordan L. Schnell[1,2,*], Vaishali Naik[1], Larry W. Horowitz[1], Fabien Paulot[3], Jingqiu Mao[1,2,**], Paul Ginoux[1], Ming Zhao[1], and Kirpa Ram[4]

[1]NOAA Geophysical Fluid Dynamics Laboratory, Princeton, New Jersey, USA
[2]Program in Atmospheric and Oceanic Sciences, Princeton University, Princeton, New Jersey, USA
[3]Cooperative Institute for Climate Science, Princeton University, New Jersey, USA
[4]Institute of Environment and Sustainable Development, Banaras Hindu University, Varanasi, India
*now at Department of Earth and Planetary Sciences, Northwestern University, Evanston, IL, USA
**now at Department of Chemistry and Biochemistry, University of Alaska, Fairbanks, Fairbanks, Alaska, USA

*Correspondence to*: Jordan L. Schnell (jordan.schnell@northwestern.edu)

**Abstract.** Northern India (23° N–31° N, 68° E–90° E) is one of the most densely populated and polluted regions in world. Accurately modeling pollution in the region is difficult due to the extreme conditions with respect to emissions, meteorology, and topography, but it is paramount in order to understand how future changes in emissions and climate may alter the region's pollution regime. We evaluate the ability of a developmental version of the new-generation NOAA GFDL Atmospheric Model, version 4 (AM4) to simulate observed wintertime fine particulate matter (PM$_{2.5}$) and its relationship to meteorology over Northern India. We compare two simulations of GFDL-AM4 nudged to observed meteorology for the period 1980–2016 driven by pollutant emissions from two global inventories developed in support of the Coupled Model Intercomparison Project Phases 5 (CMIP5) and 6 (CMIP6), and compare results with ground-based observations from India's Central Pollution Control Board (CPCB) for the period 1 October 2015–31 March 2016. Overall, our results indicate that the simulation with CMIP6 emissions produces improved concentrations of pollutants over the region relative to the CMIP5-driven simulation.

While the particulate concentrations simulated by AM4 are biased low overall, the model generally simulates the magnitude and daily variability of observed total PM$_{2.5}$. Nitrate and organic matter are the primary components of PM$_{2.5}$ over Northern India in the model. On the basis of correlations of the individual model components with total observed PM$_{2.5}$ and correlations between the two simulations, meteorology is the primary driver of daily variability. The model correctly reproduces the shape and magnitude of the seasonal cycle of PM$_{2.5}$, but the simulated diurnal cycle misses the early evening rise and secondary maximum found in the observations. Observed PM$_{2.5}$ abundances are by far the highest within the densely populated Indo-Gangetic Plain, where they are closely related to boundary layer meteorology, specifically relative humidity, wind speed, boundary layer height, and inversion strength. The GFDL AM4 model reproduces the overall observed pollution gradient over Northern India as well as the strength of the meteorology–PM$_{2.5}$ relationship in most locations.



## 1 Introduction

Air pollution in India has become a serious problem in recent years, with particulate matter of aerodynamic diameter less than 2.5 µm ($PM_{2.5}$) accounting for over 1 million premature deaths in 2015 (Health Effects Institute, 2017). The Indo-Gangetic Plain (IGP) is an extremely densely populated area in Northern India that experiences some of the highest $PM_{2.5}$

levels in the world. Considering that there are limited measurements of the spatiotemporal distribution of $PM_{2.5}$ and its aerosol components over this region, we must rely on chemistry-climate models to investigate their abundances, long-range transport, trends and variability, and predict their distributions in the future with changing climate and emissions. Here we evaluate the ability of a state-of-the-art global chemistry-climate model to reproduce surface $PM_{2.5}$ abundances using recently available hourly surface observations covering a large portion of Northern India over an entire extended winter

season (October 2015–March 2016).

Simulating $PM_{2.5}$ is a difficult task since it entails correctly modeling its individual components, which requires an accurate representation of emissions, chemistry, and transport. The task is especially challenging for simulating the distribution of $PM_{2.5}$ over Northern India due to its extreme physical (i.e., complex topography), chemical (concentrated and abundant

primary and precursor emissions), as well as dynamical meteorological (i.e., shallow boundary layer heights with frequent inversions during winter) conditions. While summer months are characterized by the south-west monsoon and relatively low pollution levels, the region's wintertime meteorology greatly favors $PM_{2.5}$ buildup; i.e., low wind speeds, shallow boundary layer heights, and high relative humidity (e.g., Nair et al., 2007). Additionally, multiple feedback mechanisms between meteorology and surface $PM_{2.5}$ exist such that, if they are not adequately represented, can amplify problems with a given

model. For example, high aerosol loading at the surface, particularly absorbing aerosols such as black carbon, can cause surface cooling and warming aloft, which can lead to an enhanced boundary layer inversion that allows greater aerosol accumulation (Ackerman et al., 2000; Ramanathan et al., 2005; Gao et al., 2015; Yang et al., 2016). The stabilization of the boundary layer caused by high aerosol concentrations can also lead to reduced wind speeds and thus decreased ventilation, as well as increased relative humidity and thus increased hydroscopic aerosol growth (Ram et al., 2014; Chen et al., 2017).

The dominant source of emissions of primary $PM_{2.5}$ and of precursors of secondary $PM_{2.5}$ in the IGP originate from coal-fired power plants and brick-kiln industries scattered throughout the region (Prasad et al., 2006) with other majors sources including agricultural biomass burning, transportation, and burning of biofuels used for heating and cooking (e.g., Reddy and Venkataraman, 2002). Some sources, primarily crop residue burning, are less steady than those from other anthropogenic

sources like power generation, but can intermittently drastically impact air quality. While the sources of these emissions are relatively well-known, there are large uncertainties in emission estimates across inventories (e.g., Jena et al., 2015; Zhong et al., 2016). Indeed, correctly simulating the extremely high $PM_{2.5}$ abundances in the IGP has proved troublesome for current global chemistry models (Reddy et al., 2004; Chin et al., 2009; Ganguly et al., 2009; Menon et al., 2010; Henriksson et al.,



2011; Goto et al., 2011; Nair et al., 2012; Cherian et al., 2013; Moorthy et al., 2013; Sanap et al., 2014; Pan et al., 2015). A multi-model evaluation by Pan et al. (2015) concluded that an underestimation of wintertime biofuel emissions was the dominant cause of the models' low biases. With model simulations for the upcoming CoupledAerosol Chemistry Model Intercomparison Project (AerChemMIP) endorsed by the Coupled Model Intercomparison Project Phase 6 (CMIP6) in support of the sixth IPCC assessment report (AR6) presently beginning, a standing question is whether the newly updated emissions will remedy these frequently found low biases. For this analysis, we use two different emission datasets, those developed for the CMIP5 (Lamarque et al., 2010) and the CMIP6 (Hoesly et al., 2017; van Marle et al., 2017) to test their impact on modeled $PM_{2.5}$. We also assess the model's ability to reproduce the observed relationships between site level $PM_{2.5}$ abundances and meteorology. We then show which meteorological indicators have consistency in their ability to predict $PM_{2.5}$ over the past few decades so that the effect of future potential changes in meteorology and their impact on $PM_{2.5}$ abundances can be assessed. The paper is organized as follows: in Section 2 we describe the observational datasets and the model used in this study; Section 3 describes the results, and our conclusions and discussion are in Section 4.

## 2 Materials and Methods

### 2.1 Observations of surface $PM_{2.5}$

We use surface observations of hourly $PM_{2.5}$ abundances ($\mu g\ m^{-3}$) provided by India's Central Pollution Control Board (CPCB; http://www.cpcb.gov.in/CAAQM/mapPage/frmindiamap.aspx). We consider the time period 1 October 2015–31 March 2016 since the highest $PM_{2.5}$ abundances in Northern India typically occur during late fall to early spring (Moorthy et al., 2013) and very few observations are available for years prior to this period. A total of 22 sites across Northern India provide data for this time period; however many of these sites span areas of only a few 10s of km (e.g., 9 are located in a single area of about 25 x 25 km in and around New Delhi). Table 1 provides a summary of the sites used. The data contains obvious repetitive "fill" values when presumably the monitor obtains a null value or the measurement is outside of the detectable range (e.g., 985 $\mu g\ m^{-3}$, 1,985 $\mu g\ m^{-3}$). However, these values are not flagged and are also considered in the daily averages that CPCB provides. Most of these fill values are well over 1,000 $\mu g\ m^{-3}$ and so we exclude any $PM_{2.5}$ abundance $\geq$ 985 $\mu g\ m^{-3}$ although such high levels have occasionally been observed in similarly polluted environments (e.g., Liu et al., 2017). This process filters 812 observations, or ~1% of the total hourly data. Daily averages are calculated from each site's hourly abundances as long as there is at least one valid value.

We also use surface observations of total $PM_{2.5}$ and its chemical composition collected at Kanpur (26.5° N, 80.3° E, 142 m asl) over the period 25 October 2008–30 January 2009. The chemical composition data includes organic carbon (OC) converted to OM using a factor of 1.6, elemental carbon (EC), ammonium ($NH_4^+$), sulfate ($SO_4^{2-}$), nitrate ($NO_3^-$), Na$^+$, K$^+$, Mg$^{2+}$, Ca$^{2+}$, Cl$^-$, and HCO$_3^-$. A thorough description of the sampling details and measurements is provided by Ram and Sarin (2011) and Ram et al. (2012; 2014).



## 2.2 Meteorological reanalysis data

We use reanalysis fields (2.5° x 2.5° linearly interpolated to 1° x 1°) from the National Centers for Environmental Prediction (NCEP/NCAR)(version 1; http://www.esrl.noaa.gov/psd/) of several variables known to influence $PM_{2.5}$ abundances including: relative humidity (RH), surface (2 m) temperature, precipitation, and wind speed and direction at the surface (10-m), 850 mb, and 500 mb, each of which are provided at 6-hourly intervals. We also use boundary layer height (BLH) reanalysis from ECMWF ERA-Interim (Dee et al., 2011) since NCEP/NCAR does not provide this variable; we note that there is uncertainty in BLH from reanalysis data. We calculate an inversion strength (INV) as the difference between the 850 mb and surface (2m) temperature (Gutiérrez et al., 2013).

We additionally derive metrics from the reanalysis data that relate to air stagnation, since stagnation describes the basic meteorological conditions that are thought to exacerbate the worst pollution levels (e.g., Jacob and Winner, 2009; Fiore et al., 2012; 2015; and references therein). Stagnation has also been used as a proxy for air quality under future climate change (e.g., Horton et al., 2014). The first metric we consider is the modified air stagnation index (ASI) (Wang and Angel, 1999; Horton et al., 2012), which is a Boolean index that is true (i.e., the air mass is considered stagnant) when: (i) daily average 10-m wind speed < 3.2 m s$^{-1}$, (ii) daily average 500 mb wind speed < 13.0 m s$^{-1}$, and (iii) daily total precipitation is less than < 1.0 mm. We also calculate the wind run and recirculation factor, which respectively characterize stagnation and recirculation of surface level flow as described in detail by Allwine and Whiteman, (1994). While the wind run and recirculation factor do not utilize upper level wind data like the ASI, their advantage is that they provide a measure of the magnitude of stagnation. The recirculation factor takes values between 0 and 1, with 1 meaning total recirculation of an air parcel and 0 meaning complete ventilation. The wind run is a measure of cumulative wind with a value of 276.5 km equivalent to the daily 3.2 m s$^{-1}$ cutoff for the 10-m wind speed.

## 2.3 GFDL-AM4

We use a developmental version of the new-generation NOAA Geophysical Fluid Dynamics Laboratory Atmospheric Model, version 4 (GFDL AM4) for our analysis (Zhao et al., 2017ab). The standard model setup as described by Zhao et al. consists of a cubed sphere finite-volume dynamical core with a horizontal resolution of ~100 km on a cubed sphere grid (96x96 grid boxes per cube face) with 33 levels extending from the surface up to about 50 km (1 hPa).

The physical atmospheric model (AM4) differs from the previous version (AM3) as described by Zhao et al. (2017b). AM4 contains increased resolution compared to AM3 (~200 km). AM4 has significantly updated the GFDL radiative transfer code, with refitting to line-by-line calculations using the latest spectroscopy and adding 10 micron $CO_2$ bands, among other changes. The shortwave radiation time-step is reduced from 3 hours (AM3) to 1 hour. In addition AM4 uses a new topographic gravity wave drag parameterization described by Garner (2005) and a new double plume moist convection





scheme developed based on the University of Washington shallow cumulus scheme (Bretherton et al., 2004). The new scheme represents shallow and deep convection, with stronger (weaker) lateral mixing into the shallow (deep plume), with convective inhibition closure for the shallow plume mass flux and cloud work function relaxation closure for the deep plume. The deep plume lateral mixing is also prescribed to vary with the environmental relative humidity.

The base version of AM4 described by Zhao et al. includes interactive aerosols, but only "simple" chemistry needed to drive aerosol formation by oxidation of precursors. We modify this configuration by adding detailed tropospheric and stratospheric chemistry, and expand the vertical extent of the model to 49 levels (extending up to ~80 km or 1 Pa). This vertical resolution is similar to that in AM3, except an extra layer is placed near the surface so that the model's lowermost layer is thinner.

The chemistry and aerosol physics module in our "full chemistry" version of AM4 are similar in structure to AM3, but with significant modifications. In particular, the efficiency of removal of tracers by convective precipitation is significantly increased (Paulot et al., 2016) compared to the unrealistic weak removal in AM3 (Fang et al., 2011), while the wet removal by frozen precipitation produced by the Bergeron process is strongly reduced (Liu et al., 2011; Paulot et al., 2017). As a

result, the spatial distribution of aerosol climatology and seasonal cycle simulated by AM4 has been significantly improved relative to AM3.

The base chemical mechanism used is from AM3 as described by Naik et al. (2013), with gas-phase and heterogeneous chemistry updates from Mao et al. (2013a, b), and updates to the treatment of sulfate and nitrate chemistry and revised

treatment of wet deposition (Paulot et al. 2016; 2017). Here we do not take into account the uptake of $SO_2$ and $HNO_3$ on dust. The heterogeneous uptake coefficients used here are provided in Table S1. The model includes the FAST-JX version 7.1 (Li et al., 2016) photolysis code and interactive biogenic isoprene emissions following Guenther et al. (2006), as implemented by Li et al. (2016).

Modeled aerosol species include black carbon (BC), primary organic matter (OM), anthropogenic secondary organic aerosol (SOA), ammonium ($NH_4^+$), sulfate ($SO_4^{2-}$), nitrate, ($NO_3^-$), sea salt, and mineral dust. Hydrophobic BC and OM are converted to hydrophilic with a 1.44 days e-folding time. Sea salt and mineral dust are partitioned into 5 size bins with constant volume size bin distributions. The thermodynamic equilibrium of the $SO_4^{2-}-NO_3^--NH_4^+$ system is simulated using ISORROPIA (Fountoukis and Nenes, 2007) with the equilibrium between gas and aerosol assumed to be reached at each model time step

(30 min). Modeled dry $PM_{2.5}$ abundances are defined using Eq. (1). We consider 100% of each aerosol species to be included in $PM_{2.5}$ except for sea salt and mineral dust (only the fraction of the bin with diameter < 2.5 µm is used). We calculate the dry mass of $PM_{2.5}$ as:

$$PM_{2.5}(dry) = SOA + dust1 + 0.25dust2 + ssalt1 + ssalt2 + 0.167ssalt3 + BC + OM + NH_4^+ + NO_3^- + SO_4^{2-}, \quad (1)$$



This formulation (Eq. (1)) does not consider the mass of aerosol water (i.e., hygroscopic growth). Due to diagnostic limitations, assumptions of the $SO_4^{2-}$–$NO_3^-$–$NH_4^+$ system must be made in terms of partitioning $NH_4^+$ between $NO_3^-$ and $SO_4^{2-}$ in order to retrieve hourly abundances of PM$_{2.5}$ that includes hygroscopic growth, since the growth factors are described for

$NH_4NO_3$ and $(NH_4)_2SO_4$. At each hourly time step, we calculate the fraction (α) of $NH_4^+$ present as $(NH_4)_2SO_4$ using:

$$\alpha = 2SO_4^{2-}/(NO_3^- + 2SO_4^{2-}) \, , \qquad (2)$$

The remaining $NH_4^+$ forms $NH_4NO_3$. We partition $NH_4^+$ this way since $(NH_4)_2SO_4$ is more stable. The $(NH_4)_2SO_4$ hygroscopic

growth factor is then applied to $SO_4^{2-}+\propto NH_4^+$ and thus the $NH_4NO_3$ growth factor is applied to $NO_3^- + (1-\propto)NH_4^+$. We recognize using this method (as opposed to online calculation using ISORROPIA) introduces some uncertainty into the modeled aerosol mass. However, we use this method since we want to apply constant hygroscopic growth factors at 50% RH in order to realistically compare modeled results to the observations, as this RH value is operationally defined by CPCB. In addition, there are variations in how different models represent hygroscopic growth. For example, the GEOS-CHEM

chemical transport model (Martin et al., 2003) applies the same hygroscopic mass growth factor at 50% RH of 1.51 to $NH_4^+$, $NO_3^-$, and $SO_4^{2-}$, which is slightly higher than our growth factors for $NH_4NO_3$ (1.32) and $(NH_4)_2SO_4$ (1.46). In addition, GEOS-CHEM includes a growth factor of 1.24 for hydrophilic OM and SOA whereas our parameterization has no hygroscopic growth of organics at 50% RH (Ming and Russel, 2004). Sea salt has the largest growth factor (2.32). In any case, we present both dry and wet aerosol mass whenever possible in order to highlight the uncertainty involved with aerosol

water content. Equation (3) shows the calculation of PM$_{2.5}$ mass including water (hence wet PM$_{2.5}$).

$$PM_{2.5}(\text{wet @50\%RH}) = SOA + dust1 + 0.25dust2 + 2.32(ssalt1 + ssalt2 + 0.167ssalt3) + BC + OM + 1.32(NO_3^- + (1-\propto)NH_4^+) + 1.46(SO_4^{2-}+\propto NH_4^+) \, , \qquad (3)$$

The model is nudged with NCEP-NCAR reanalysis winds (Lin et al., 2012) using a pressure-dependent nudging technique and a relaxation time scale of 6 h at the surface and weakening to ~60 h by 100 ha; this facilitates a consistent comparison of modeled and observed daily PM$_{2.5}$ abundances.

We perform two simulations, each of 36 years and 3 months duration (1 Jan 1980 – 31 March 2016), with the first year used

as model spin-up. The first simulation, denoted as AM4-CMIP5, is driven by emissions developed in support of the CMIP5 (Lamarque et al., 2010) and is extended from 2000 to 2016 following the Representative Concentration Pathway 8.5 (RCP8.5; van Vuuren et al., 2011). The second simulation, referred to as AM4-CMIP6, is driven by emissions developed for the upcoming CMIP6 (Hoesly et al., 2017; van Marle et al., 2017) wherein the period from 2000 to 2014 represents historical emissions rather than projections. Years 2015-2016 emissions were set to year 2014 values.





The spatial patterns (1° x 1°) of the CMIP6 total anthropogenic extended wintertime (October–March, 2015–2016) emissions (Gg) of BC, OM, NO, $SO_2$, and $NH_3$ used over India in the AM4-CMIP6 simulation are shown in Fig. 1a-e, respectively. Emissions are largest for all species over the IGP and lowest over the oceans and the Himalayan Plateau. The
patterns of BC, OM, $NO_2$, and $NH_3$ are fairly uniform over the IGP while $SO_2$ emissions (Fig. 1d) are highly localized due to large emissions from coal-fired power plants.

Figure 1f-i show the percent difference between CMIP6 and CMIP5 emissions for the same time period and for the five species in Fig. 1a-e. For BC, OM, NO, and $NH_3$ emissions, most grid cells within the IGP show differences of 50–100%.
Differences in $SO_2$ emissions are more variable, with some grid cells showing moderate differences while others greater than 100%. Figures S1-S5 in the Supplement compare the evolution of these emissions over the full time period of the model simulation for ten sub-regions in India. For almost every region and every species, the difference between CMIP5 and CMIP6 emissions is largest over 2000–2016 period (i.e., when CMIP5 is extended using RCP8.5), however, some large differences are also seen in earlier years. Overall, anthropogenic emissions in the CMIP6 dataset are much higher than those
in the CMIP5 dataset.

## 3 Results

In this section we use the surface measurements of $PM_{2.5}$ to evaluate the ability of GFDL-AM4 to reproduce the daily concentrations of $PM_{2.5}$ and their relationship with meteorological variables, large-scale percentile patterns, and diurnal and seasonal cycles.

### 3.1 Daily Variability

Figure 2a-b show the 6-month time series of daily average $PM_{2.5}$ for two grid cells with multiple observation sites. Nine observations sites are located in the grid cell over New Delhi (Fig. 2a), and they show extreme variability between them despite being within ~25 km of one another. The average difference among the maximum and minimum abundance of the sites on a given day is 160 µg m$^{-3}$, sometimes reaching over 400 µg m$^{-3}$. The AM4-CMIP5 abundances are a factor of ~2–4
less than the observations, but the actual magnitude of the bias clearly depends on which observation is being compared against. The AM4-CMIP6 abundances are also generally biased low, mainly in December–January, but often fall within the range of the observations for other months. The correlation coefficient (r) between the two simulations is 0.96 and 0.94 for New Delhi and Kanpur/Lucknow, respectively, highlighting the strong control of meteorology on daily variability. The correlation of the AM4-CMIP6 dry and wet $PM_{2.5}$ with the average of the observations is 0.57 and 0.56, respectively. The
four sites located in the grid cell over the cities of Kanpur and Lucknow (~75 km apart)(Fig. 2b) show less variability between them than those over New Delhi, with an average min-to-max difference of 94 µg m$^{-3}$. The correlation of AM4-





CMIP6 with the average of the observations is slightly better for this grid cell, 0.65 (dry) and 0.63 (wet). In both grid cells, the inclusion of aerosol water slightly decreases the correlation between models and average observations, however, it reduces the normalized mean bias (NMB) from –41% to –28% and –26% to –10% over New Delhi and Lucknow/Kanpur, respectively. This is true for most of the individual sites as well: only four sites show an improvement in correlation and all but two sites show a reduction in the NMB  (Table S2). Noteworthy is the model bias for both locations during the 5-day festival of Diwali (shown by the vertical dashed lines in Fig. 2), which is a period marked with extensive fireworks celebrations and thus high aerosol loading (Tiwari et al., 2012) that the model obviously would not reproduce since the emission inventories used here have monthly resolution. These plots show that $PM_{2.5}$ abundances can vary substantially over regions smaller than a model grid cell (i.e., ~100 km), especially in megacities such as New Delhi (~22 million people) and Lucknow/Kanpur (~6 million people). The model abundances represent an average over the grid cell while the observations represent a point measurement, and thus this incommensurability necessitates caution when comparing the two (e.g., Schnell et al., 2014).

Figure S6a-d show the same time series for the observations and both simulations but separates modeled total $PM_{2.5}$ into its individual components. For both grid cells and simulations, $NO_3^-$ is the dominant driver of modeled $PM_{2.5}$ variability, accounting on average around one-third of $PM_{2.5}$, largest on days with the highest total $PM_{2.5}$. Thus, the model's ability to match the time series of PM2.5 observations is largely dependent on $NO_3^-$ variability. OM is the second largest contributor to total $PM_{2.5}$ (~25%), but is less variable than $NO_3^-$.  Table S2 also shows the correlation of each site's observed PM2.5 time series with each of the modeled components of $PM_{2.5}$. For most sites, the correlation with either OM or BC is highest (shown by the bolded numbers).

We next compare the fractional contribution of each $PM_{2.5}$ component to the speciation measurements in Kanpur during October 2008– January 2009. The observations include an "unidentified" component so we separately show the observed fractions excluding the unidentified fraction (Fig. 3a) and including it (Fig. 3b). The results for AM4-CMIP5 and AM4-CMIP6 are shown in Fig. 3c and Fig. 3d, respectively.  The average total $PM_{2.5}$ abundance is shown below each pie chart. The largest model-measurement discrepancy is for the OM and $NO_3^-$ component fractions. The model underestimates the OM contribution by 12–27% for AM4-CMIP5 and 19–34% for AM4-CMIP6, depending on whether the unidentified fraction is included. The model overestimates the $NO_3^-$ contribution by 19-21% for AM4-CMIP5 and 30–32% for AM4-CMIP6, again depending on whether the unidentified fraction is included. In absolute terms, the AM4-CMIP5 and AM4-CMIP6 is overestimating $NO_3^-$ by 17 µg m$^{-3}$ (217%) and 53 µg m$^{-3}$ (680%), respectively. For OM, AM4-CMIP5 underestimates it by 6 µg m$^{-3}$ (-15%) while AM4-CMIP6 overestimates it by 7 µg m$^{-3}$ (18%). The unidentified fraction is quite large (42% of total $PM_{2.5}$, Fig. 3b), which Ram and Sarin (2011) hypothesize to be mineral dust; while the AM4 may underestimate wintertime dust like previous versions of the model (Ganguly et al., 2009), it is unlikely the underestimate is this large.



## 3.2 Spatial patterns of PM$_{2.5}$ and related meteorology

Figure 4a-c respectively show the 5th, 50th, and 95th percentile of the AM4-CMIP6 (wet) daily average PM$_{2.5}$ abundances for 1 October 2015 – 31 March 2016 overlain with the observed site values over the same time period. As expected from the emission patterns in Fig. 1, the highest PM$_{2.5}$ abundances in northern India are found throughout the IGP for all percentiles. Within the IGP, the eastern edge shows the highest abundances in both observations and AM4 simulations. This feature is most evident at the 95th percentile, where the observed values in the east exceed those in the north and central IGP by up to 50%, with the 95th percentile value at one site reaching ~400 µg m$^{-3}$ – 16 times the WHO-recommended health standard for a 24-hour average abundance (WHO, 2006). The AM4-CMIP6 simulation is biased slightly low at the 50th (normalized mean bias (NMB) = –19%) and 95th percentiles (NMB = –22%) for most sites, while slightly high for some sites at the 5th percentile (NMB = +4%). This is a major improvement compared to the AM4-CMIP5 (wet) simulation (Fig. S7), which has NMBs at the 5th, 50th, and 95th percentile of –35%, –53%, and –57%, respectively. The changes from CMIP5 to CMIP6 are almost entirely in terms of magnitude as the spatial correlations between the AM4-CMIP5 and AM4-CMIP6 percentile maps are all around $r = 0.99$.

Figure 5 shows modeled (AM4-CMIP6) and observed RH, BLH, INV, and 10-m, 850 mb, and 500 mb wind speed and direction averaged over the 1 October 2015–31 March 2016 time period. Note that the reference wind vectors in Figs. 5d and 5f are the ASI cutoffs for 10-m and 500 mb wind speeds, respectively. Overall, the AM4 simulates these meteorological quantities reasonably well, as should be expected since the wind fields in AM4 are nudged to reanalysis data. Ideally, these comparisons should be made with surface observations and soundings, however data limitations hinder a complete evaluation of modeled meteorology.

The general northwesterly direction of the 10-m and 850 mb winds (generally biased high for 850mb) along the IGP (Fig. 5d-e) allows for accumulation of aerosols as air masses flow to the southeast across the high-emission IGP region (e.g., Nair et al., 2007; Kumar et al., 2015; Sen et al., 2017). Nair et al. (2007) highlights the role of this transport mechanism by showing PM$_{2.5}$ levels in the IGP increased with the distance the air mass had traveled from the west. Nair et al. (2007) note that this also applies to the transport of weather phenomena conducive to aerosol buildup such as a cold air mass. The surface winds are light and variable at the eastern edge of the IGP, a recirculation pattern that inhibits outflow into the Bay of Bengal. The winds at 500mb are strong westerlies with the largest values (~18 m s$^{-1}$) in the central and eastern IGP.

The modeled and observed correlations of daily-average PM$_{2.5}$ with the meteorological variables in Fig.5a-f is shown in Fig 5g-l, respectively. Stippling and circles with an 'x', respectively, denote modeled and observed correlations significant at the 95% confidence based on a Student's t-test. Overall, AM4 reproduces most of the observed PM$_{2.5}$–meteorology correlation patterns well, especially in the eastern IGP, which consistently shows the strongest magnitude correlations for most





variables. As for the meteorological variables, INV and BLH generally show the strongest correlations ($|r| > 0.6$ in the eastern IGP). RH is positively correlated with $PM_{2.5}$ over all of northern India except for the western edge bordering the coast. Small negative correlations are found for 850mb winds while, unexpectedly, positive correlations are found between $PM_{2.5}$ and 500 mb wind speed. $NO_3^-$ is the dominant driver of this positive correlation with 500 mb winds (see below; Fig.

S8), but all anthropogenic aerosol components show high positive correlation over the far eastern edge. The negative correlation of $PM_{2.5}$ with surface and 850 mb winds and positive correlation with 500 mb winds suggest that the surface and upper-level airflow are independent. This finding is explored further in Sect. 3.3 in the context of the ASI.

The correlations of the meteorological variables with the components of $PM_{2.5}$ are not always comparable to those with total

$PM_{2.5}$. Figure S8 shows the correlations with each of the main components of $PM_{2.5}$ (i.e., dust, sea salt, BC, OM, SOA, $NH_4^+$, $SO_4^{2-}$, and $NO_3^-$) with the same meteorological variables as in Fig. 5. The correlations in Fig. S8 are computed over 2011-2015 to reduce the influence of interannual variability. For most variables and regions, particularly in the IGP, the individual $PM_{2.5}$ components have the same sign and similar magnitude correlations as those for total $PM_{2.5}$. Because $NO_3^-$ is the dominant component of $PM_{2.5}$ in most regions (Fig. S9), the correlations for $NO_3^-$ are very similar to those for total $PM_{2.5}$.

But, in some cases, differences reflect the source of the component (e.g., dust vs. BC) and aerosol chemistry. At some locations, dust and sea salt have several correlations that are opposite in sign than the rest of the components. The positive correlations with surface wind are expected given that emissions of dust and sea salt are simulated as a function of wind speed (Ginoux et al., 2001), which is especially evident near strong source regions such as the western coast (sea salt) and the Thar Desert (dust).

Since both the modeled and observed $PM_{2.5}$ reflect constant 50% RH conditions, the correlation of RH and $PM_{2.5}$ is not due to hygroscopic growth. In Fig. S8 the correlation with RH is positive for all $PM_{2.5}$ components except dust. Since $NO_3^-$ and OM are the dominant components of total $PM_{2.5}$ over most locations (Fig. S9), their correlations with RH (and thus the other meteorological variables) are what is predominantly reflected in terms of total $PM_{2.5}$. We also examine the correlation of RH

with the other meteorological variables to determine the underlying driver of the correlations of RH and $PM_{2.5}$ and its components. RH has large negative correlations with surface wind speeds and is positively correlated with INV and low cloud cover (not shown), which all are indicative of a stable boundary layer (e.g. Chen et al., 2017) and would cause $PM_{2.5}$ to accumulate (e.g., high RH ~ low wind speeds, low ventilation → high $PM_{2.5}$). Essentially, the positive correlation of $PM_{2.5}$ with RH likely reflects that high RH is indicative of other meteorological conditions that allow $PM_{2.5}$ to accumulate. For

$SO_4^{2-}$, the positive RH relationship may also reflect in-cloud $SO_2$ oxidation (e.g., Ram et al., 2012).




### 3.3 Seasonal and diurnal cycles

The seasonal cycle of total $PM_{2.5}$ over the IGP is driven by a combination of meteorology and emissions. For example, the relatively cold winters in the IGP (~7°C average daily December–January minimum over Kanpur/Lucknow) increases overall energy demand and leads to additional burning of fuel for indoor heating. The colder temperatures also play a role in
the chemistry of $PM_{2.5}$ (e.g., nitrate is more stable at colder temperatures). The strong radiative cooling at the surface during the winter nights often result in foggy conditions and a shallow inversion layer that traps aerosols near the surface.

Figure 6a shows the monthly average $PM_{2.5}$ abundances for the min-to-max range of the individual stations (gray shading) and their median (black line) over the 6-month period. Also shown is the median of modeled $PM_{2.5}$ for AM4-CMIP5 dry
(blue), AM4-CMIP6 dry (red), AM4-CMIP6 wet (green). The maximum $PM_{2.5}$ abundances in the observations occur over a broad peak from November–January, with a mean of ~200 µg m$^{-3}$ and individual sites ranging from ~125–300 µg m$^{-3}$. Both AM4 simulations reproduce month-to-month variations well, but the AM4-CMIP5, although biased very low, matches the January peak that the AM4-CMIP6 misses.

Figure 6b-e show the annual cycle for RH, BLH, INV, and 10-m wind speeds, respectively. As expected from the correlations shown in Fig. 5g-l, the cycles of these variables align, such that meteorology is most conducive to formation and accumulation during the months with maximum $PM_{2.5}$: e.g., low BLH and strong INV trap pollution near the surface; and low wind speeds decrease ventilation. High RH also increases ambient observed $PM_{2.5}$ due to hygroscopic growth, but the connection here is probably due to the relationship between RH and a stable boundary layer. The model reproduces the shape
of these cycles well, but is biased high in RH (~15% from November to January), low in BLH (~200 m for all months), high in INV (~2° for most months), and low in surface wind speed (0.5 to 1 m s$^{-1}$). There is some evidence for aerosol feedback onto meteorology seen by comparing the curves of AM4-CMIP5 (blue) to AM4-CMIP6 (green), namely a stabilization effect on the boundary layer. The much higher aerosol abundances in AM4-CMIP6 seem to have caused higher relative humidity, decreases in boundary layer height and surface wind speeds, as well as a stronger inversion.

The skill of the model in reproducing the seasonal cycle of $PM_{2.5}$ and related meteorology provides confidence in the seasonality of emissions and the ability of the model to simulate the large-scale $PM_{2.5}$-meteorology relationships. A more stringent test is to compare the diurnal cycles from observations and the model. Like the seasonal cycle, the diurnal cycle of $PM_{2.5}$ is controlled by a combination of emissions and meteorology. However, since the model lacks a diurnal cycle of
emissions, it may not be capable of accurately reproducing the observed diurnal cycle of $PM_{2.5}$. If the model were to simulate the observed diurnal cycle well, even without a diurnal cycle of emissions, this would provide confidence in the model's ability to simulate high-frequency $PM_{2.5}$–meteorological relationships.



Figure 6f shows the diurnal cycle of $PM_{2.5}$ for the min-to-max range of the individual stations (gray) and their average (black) over the 6-month period. The AM4-CMIP5 (dry), AM4-CMIP6 (dry), and AM4-CMIP6 (wet) values are also shown in blue, red, and green, respectively. We also show the diurnal cycle of each major $PM_{2.5}$ component for AM4-CMIP6 (dry) in Figure S10 of the Supplement. The minimum $PM_{2.5}$ abundance in the observations occurs at ~15:00 LT, aligning with the

lowest RH, deepest boundary layer, and highest wind speeds. As evening approaches, wind speeds decrease and the boundary layer collapses, trapping the $PM_{2.5}$ emitted and produced over the day at the surface. These features, together with an evening pulse in traffic and biofuel emissions for heating and cooking (Rehman et al., 2011), cause $PM_{2.5}$ abundances to reach their maximum at around 21:00 LT. $PM_{2.5}$ abundances decrease through the night, likely due to a drop off in emissions since all other meteorological variables would suggest further $PM_{2.5}$ increases. Abundances begin to increase about an hour

before the ~07:00 sunrise and a short-lived secondary maximum occurs at around 09:00 LT. Part of this increase may be driven by emissions (e.g., cold engine starts and morning traffic); however, Nair et al. (2007) attribute the rise mostly to fumigation (Stull, 1988; Fochesatto et al., 2001), i.e., thermals that break up the nighttime inversion layer and mix aerosols trapped in the residual layer down to the surface.

The $PM_{2.5}$ concentrations simulated by AM4 are biased low during all hours of the day, with the largest bias at the time of the evening maximum. AM4 reproduces a slightly delayed morning maximum, closely following the rise in BC and OM/SOA (Fig. S10b-c). AM4 largely misses the evening rise in $PM_{2.5}$ abundances despite simulating the changes in meteorology, which would otherwise increase $PM_{2.5}$ abundances. Thus, the lack of a diurnal emission cycle in the AM4 likely explains why it misses the evening peak in $PM_{2.5}$. Indeed, Ram and Sarin (2011) find that, for a site in Kanpur,

boundary layer dynamics are not the only cause for the evening rise $PM_{2.5}$ but rather an increase in source emissions and enhanced secondary aerosol formation. Additionally, Ram and Sarin (2011) find that particulate $NO_3^-$ was a factor of four higher during nighttime, attributable to secondary formation via the hydrolysis of $N_2O_5$ under high humidity conditions. AM4, however, shows the reverse, with a peak in $NO_3^-$ during midday. Overall, $NH_4^+$, $NO_3^-$, and $SO_4^{2-}$ in AM4 (Fig. S10d–f) act to increase midday $PM_{2.5}$ and cause a relative decrease in morning and early evening $PM_{2.5}$, thereby decreasing the

overall amplitude of the $PM_{2.5}$ diurnal cycle.

### 3.4 Air stagnation

The Air Stagnation Index (ASI) is a metric commonly used to identify days when meteorology is conducive to the buildup of pollutants, viz. light winds at the surface and upper levels and no precipitation (Wang and Angell, 1999). Meteorological, light winds at the surface hinder dilution of pollutants, no precipitation prevents pollutant washout and implies no convection

and thus less dilution, and light upper level winds are related to slow moving synoptic systems. The index was developed over the United States, but it has been used in relation to air quality in other regions (e.g., Horton et al., 2012; 2014; Huang et al., 2017). Fig. S11 shows the frequency of each of the stagnation criteria over the 6-month period. Stagnation with respect to 10-m wind speeds and precipitation occurs on nearly 100% of the days over large portions of northern India. The 500 mb



wind speed criteria is the limiting factor for total ASI, both occurring on ~35% of the days. These patterns of stagnation frequency are nearly identical from year to year over the period of the simulations, with only a few grid cells (most over the ocean and the Himalayas) showing significant trends in any of the stagnation criteria (not shown).

The large negative correlations between $PM_{2.5}$ and 10-m wind speed shown in Fig. 5j clearly imply that days with low surface wind speeds have higher $PM_{2.5}$ abundances. Precipitation is weakly correlated with $PM_{2.5}$ (not shown), but it is relatively rare during winter months. However, the positive correlation of $PM_{2.5}$ with 500 mb wind speeds – the third component of the ASI – suggests that the ASI may not accurately describe conditions susceptible to pollution buildup over India.

We test the ability of the ASI to predict extreme pollution days by comparing the composite average $PM_{2.5}$ on days when any individual or all stagnation criteria are met with the average on days that they are not met. We calculate the composite using anomalies relative to the monthly mean to remove the influence of seasonality. Notably, since nearly 100% of the days are considered stagnant with respect to 10-m wind speeds and precipitation, the composites are not a completely fair
comparison. However, the 500mb and total ASI criteria are achieved as often (Fig. S11). Figure 7a-d show the results for observations and AM4-CMIP6 over the October 2015 – March 2016 time frame. The gray regions in Fig. 7a and 7d are grid cells where a composite cannot be constructed since 100% of the days are considered stagnant for the respective criterion. Both the observations and AM4-CMIP6 have large positive composites (~30 µg m$^{-3}$ ASI minus non-ASI, relative to the monthly mean) in the IGP on days when the 10-m wind ASI criterion is met. The results for the precipitation criterion are
mixed, with the observations showing positive composite anomalies for most sites, but the model only showing positive composites for areas in the far eastern edge of the IGP. The composites for the 500 mb component and total ASI agree very well, but are also mixed between the observations and model. For the observations, most composites are slightly positive (0–10 µg m$^{-3}$) with the model showing similar magnitudes but opposite in sign. Examining the composites on a monthly basis shows that the positive composites for 500 mb and total ASI in the observations occur mostly outside of December-
January, the months when the highest $PM_{2.5}$ abundances typically occur. Indeed, plots of the composites using the raw data (i.e., not relative to the monthly mean) (Fig. S12) shows that nearly all observations and all model grid cells have large negative composites for the 500 mb wind speed and total ASI.

We test if these composites are representative of a broader time period by calculating the composites for seven 5-y intervals
over the 1981–2015 period in AM4-CMIP6 (Fig. S13). The spatial patterns of the 10-m wind and precipitation composites are extremely similar across the 5-y intervals, although the magnitude increases with time. Interestingly, the results for the 500 mb and total ASI composites in the earlier decades are different from those in the last decade. In the first two decades the composites are near zero or slightly positive for most of northern India except in the far eastern edge. The last decade, however, shows negative composites over most of the domain. Unfortunately there are no observations to confirm these



results, but it nevertheless suggests a regime change in the relationship between wintertime PM$_{2.5}$ and synoptic meteorology, possibly resulting from a combination of changing climate and emissions.

These results suggest that atmospheric conditions at the surface are decoupled with those in the upper atmosphere over Northern India. This disconnect is frequently observed in mountainous topography, such as the western US (e.g., Wolyn and Mckee, 1989; Holmes et al., 2015; Chachere and Pu, 2016) where cold air pools form at the surface, such that a temperature inversion and/or deep stable boundary layer traps pollution close to the surface (Whiteman et al., 2001), leaving light surface winds despite relatively strong winds aloft (Wolyn and Mckee, 1989). A complete decoupling between the lower and upper atmosphere would presumably yield a near zero or insignificant correlation between surface PM$_{2.5}$ and 500mb wind speeds; however significant correlations greater than +0.35 are found over the entirety of the IGP – and greater than +0.60 over the far eastern edge bordering Bangladesh. It remains unclear what – if any – mechanism is responsible for the positive relationship between PM$_{2.5}$ and 500 mb wind speeds.

**3.5 What meteorological conditions consistently result in degraded air quality?**

The future of air quality over Northern India will largely be determined by emission changes. However, even with drastic emission reductions, pollutant levels may still exceed recommended levels due to the size of the population and the socioeconomics in the region. Meteorological changes due to a warming climate will also play a role, especially for the days with the highest pollution levels. It is thus important to identify meteorological conditions that have consistent effects on air quality – at least from a model standpoint – in order to provide confidence in how air quality may respond to changes in those variables.

In Fig. 8, we show PM$_{2.5}$ and meteorological composites for the 10 days with the highest PM$_{2.5}$ abundance for each observations site and each grid cell in AM4-CMIP6-wet (~95th percentile over October 2015 – March 2016) minus the 10 days with the lowest abundances (i.e., the ~5th percentile). For PM$_{2.5}$ (Fig. 8a), abundances are up to 200 µg m$^{-3}$ higher on the most polluted days compared to the cleanest days in both the model and observations. The pattern is matched extremely well by the model despite being biased low for PM$_{2.5}$ at most locations and times. Compared to the cleanest days, the most polluted days in most locations have higher RH (Fig. 8b, +5 to +15%), lower BLH (Fig. 8c, –50 to –200 m), and stronger temperature inversions (Fig. 8d, +1 to +3 K). We also perform the composites for two other stagnation metrics, the wind run and recirculation factor (Allwine and Whiteman, 1994). The wind run (i.e., a similar measure to daily average wind speed) is much lower over the IGP (about –100 km), but over the western side of Northern India it is higher in the model (about +50 km) and lower in the observations (about –50 km). Since this area has higher dust and sea salt fractions (Fig. S9), which are both parameterized as function of wind speed, and the model does not seem to be biased high in wind speed (Fig. 5d), this may indicate the dust emission source is too large or too sensitive to wind speed. For wind recirculation, most sites have





positive composite values (i.e., increased recirculation), with the model showing positive composites along the length of the IGP and largest near the far eastern edge of the IGP bordering Bangladesh where the surface winds are variable (Fig. 5d).

We extend this analysis for seven 5-year intervals to test if these relationships have changed over the recent decades and plot the results in Fig. S14. Here we use the 50 most polluted and cleanest days over 5 years (still ~95th percentile minus the ~5th percentile). Notably, the composite values for $PM_{2.5}$ have increased substantially (~40 to >200 $\mu g\ m^{-3}$) over the period due to massive increase in emissions. Overall, however, the results are extremely similar to Figure 8, but the patterns are more pronounced than those using only one winter and 10 days. Based on the consistent results over the time period, we suggest using these variables (or possibly others unexplored here) to gauge potential future changes in poor air quality days due to
changing meteorology.

## 4 Conclusions and Discussion

We have investigated the ability of a developmental version of the new-generation NOAA Geophysical Fluid Dynamics Laboratory Atmospheric Model, version 4 (GFDL AM4) to reproduce observed $PM_{2.5}$ and its relationship to meteorology over Northern India during October–March, 2015–2016. We find the new emission dataset developed for phase 6 of the
Coupled Model Intercomparison Project (CMIP6) vastly reduces the low bias of the AM4 results, nearly doubling the amount of $PM_{2.5}$ simulated over the time period. In both the observations and the model, the highest $PM_{2.5}$ abundances are found in the Indo-Gangetic Plain (IGP), specifically in the eastern states of Uttar Pradesh and Bihar. This area is also most sensitive to meteorological variables that describe the stability of the lower atmosphere including: relative humidity, boundary layer height, strength of temperature inversion, and low level wind speed.

In the AM4, nitrate ($NO_3^-$) and organic matter (OM) are the dominant components of total $PM_{2.5}$ over most of Northern India, and they are also the most sensitive components to meteorology. OM and BC are most strongly correlated with total observed $PM_{2.5}$, likely reflecting the stronger influence of meteorology compared to chemistry. Future development of AM4 to improve its ability to reproduce observed $PM_{2.5}$ over India should focus on improving its simulation of $NO_3^-$, the most
abundant $PM_{2.5}$ component in the model, which is largely overestimated compared to limited observations. AM4 correctly simulates large-scale percentile patterns of $PM_{2.5}$ as well as the seasonal (October – March) cycle. The diurnal cycle is also simulated well, but AM4 misses the early evening rise and secondary peak found in the observations, possibly because it lacks a diurnal emission cycle.

We additionally find that the air stagnation index (ASI), a commonly used indicator of poor air quality, is generally not able to predict high pollution days in the present decade over the most polluted regions of Northern India. Results are somewhat mixed for previous decades, suggesting that the success of a particular stagnation index in indicating high pollution levels in



one climate regime does not imply it will continue to be effective in different climate regimes, even on relatively short (30 year) time scales. Instead we find that poor air quality days can be better predicted using other meteorological variables describing only the stability of the lower atmosphere (i.e., surface wind speed, boundary layer height, strength of temperature inversion), relationships that have not changed in the recent past.

This analysis is largely based on a single winter of observations over Northern India. While the results have provided valuable insight into the meteorological and chemical controls on air quality, it is imperative that long-term, reliable pollutant and meteorological measurements are maintained in the region in order to better assess the future of air quality in response to changing emissions and climate.

10 **Acknowledgements**

KR thanks Department of Science and Technology, Govt. of India for providing financial support under the INSPIRE Faculty scheme (No. DST/INSPIRE Faculty Award/2012; IFA-AES-02). The authors thank Bing Pu and Meiyun Lin for their helpful comments.

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



**Table 1.** Description of observation sites

| No. | State | City | Station | Lat. (°N) | Lon. (°E) | Elev. (m) | % days[a] |
|------|-------|------|---------|-----------|-----------|-----------|-----------|
| (1) | Bihar | Gaya | Gaya Collectorate | 24.75 | 84.94 | 111 | 59.6 |
| (2) | Bihar | Muzaffarpur | Muzaffarpur Collectorate | 26.08 | 85.51 | 60 | 87.4 |
| (3) | Bihar | Patna | IGSC Planetarium Complex | 25.36 | 85.08 | 53 | 94.5 |
| (4) | Delhi | New Delhi | Anand Vihar | 28.65 | 77.30 | 205 | 84.7 |
| (5) | Delhi | New Delhi | Dwarka | 28.61 | 77.04 | 213[b] | 80.3 |
| (6) | Delhi | New Delhi | IHBAS | 28.61 | 77.21 | 213[b] | 35.5 |
| (7) | Delhi | New Delhi | Mandir Marg | 28.64 | 77.20 | 213[b] | 89.6 |
| (8) | Delhi | New Delhi | Punjabi Bagh | 28.67 | 77.13 | 216 | 92.3 |
| (9) | Delhi | New Delhi | R K Purnam | 28.57 | 77.18 | 213[b] | 92.9 |
| (10) | Delhi | New Delhi | Shadipur | 28.65 | 77.16 | 213[b] | 98.9 |
| (11) | Gujarat | Ahmedabad | Maninagar | 23.00 | 72.60 | 53 | 20.2 |
| (12) | Haryana | Faridabad | Sector 16A Faridabad | 28.41 | 77.31 | 198 | 95.1 |
| (13) | Haryana | Gurgaon | HSPC Gurgaon | 28.45 | 77.03 | 217 | 19.7 |
| (14) | Haryana | Panchkula | Panchkula | 30.71 | 76.85 | 365 | 54.1 |
| (15) | Rajasthan | Jaipur | Jaipur | 26.97 | 75.77 | 431 | 67.8 |
| (16) | Rajasthan | Jodhpur | Jodhpur | 26.29 | 73.04 | 231 | 72.1 |
| (17) | Uttar Pradesh | Agra | Sanja Palace | 27.20 | 78.01 | 171 | 97.8 |
| (18) | Uttar Pradesh | Kanpur | Nehru Nagar | 26.47 | 80.33 | 126 | 96.2 |
| (19) | Uttar Pradesh | Lucknow | Central School | 26.85 | 81.00 | 123 | 97.3 |
| (20) | Uttar Pradesh | Lucknow | Lalbagh | 26.85 | 80.94 | 123 | 98.9 |
| (21) | Uttar Pradesh | Lucknow | Talkatora | 26.83 | 80.89 | 123 | 92.3 |
| (22) | Uttar Pradesh | Varanasi | Ardhali Bazar | 25.35 | 82.98 | 80.7 | 98.9 |

[a]Percentage of days (out of 183) with a valid value

[b]Elevation estimate





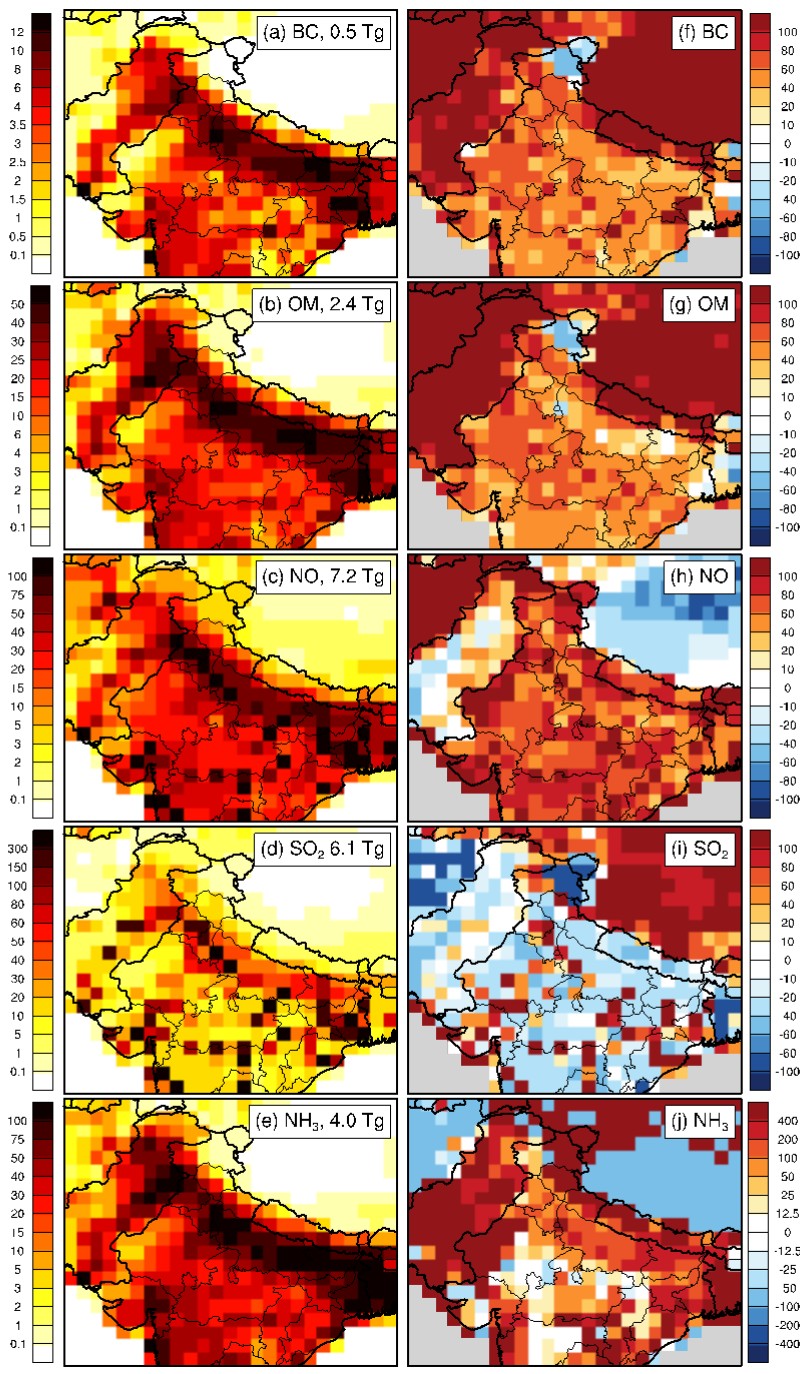

**Figure 1.** (a-e) Total CMIP6 anthropogenic emissions (Gg) of (a) black carbon (BC), (b) organic matter (OM), (c) NO₂, (d) SO₂, and (e) NH₃ over 1 October 2015 – 31 March 2016. Total emissions over the domain (in Tg) are provided in the panel titles. (f-j) Difference (%) between CMIP6 and CMIP5 emissions (CMIP6 minus CMIP5) for the species in (a-e).





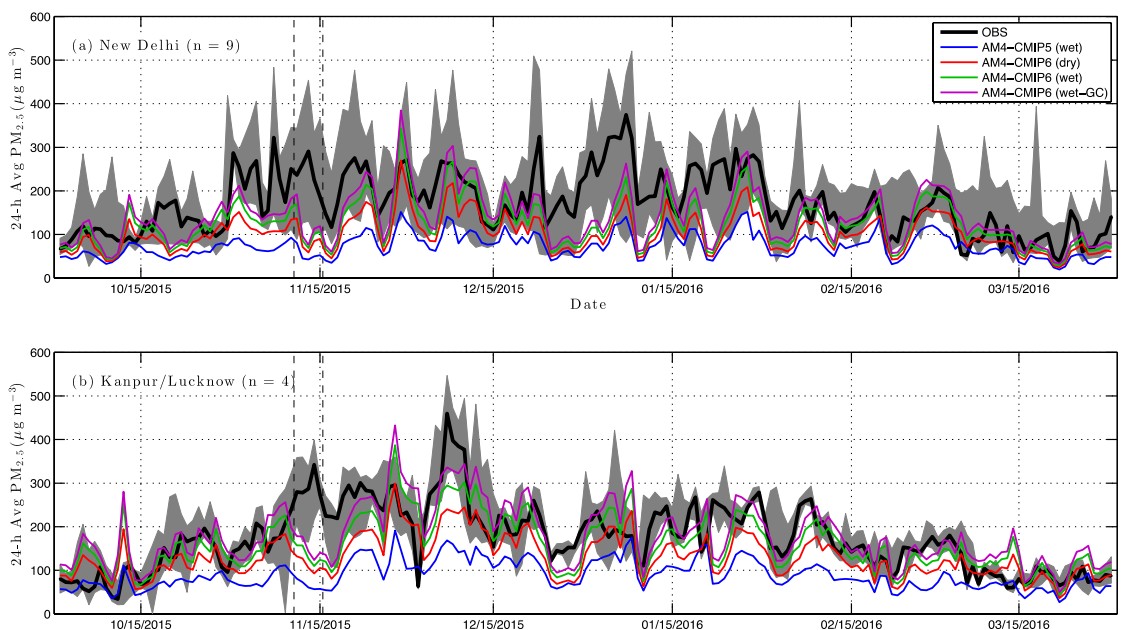

**Figure 2.** Time series of daily average PM$_{2.5}$ (1 October 2015 – 31 March 2016) for the grid cells over (a) New Delhi and (b) Kanpur/Lucknow, Uttar Pradesh. The min-to-max range of the multiple observations sites are shown in grey with their median in black. The number of sites is given in the panel titles. Modeled abundances are shown in (blue) CMIP5-dry, (red) CMIP6-dry, (green) CMIP6-wet, and (magenta) CMIP6-wet calculated with the GEOS-CHEM hygroscopic growth factors. The dashed vertical lines represent the 5-day festival of Diwali.



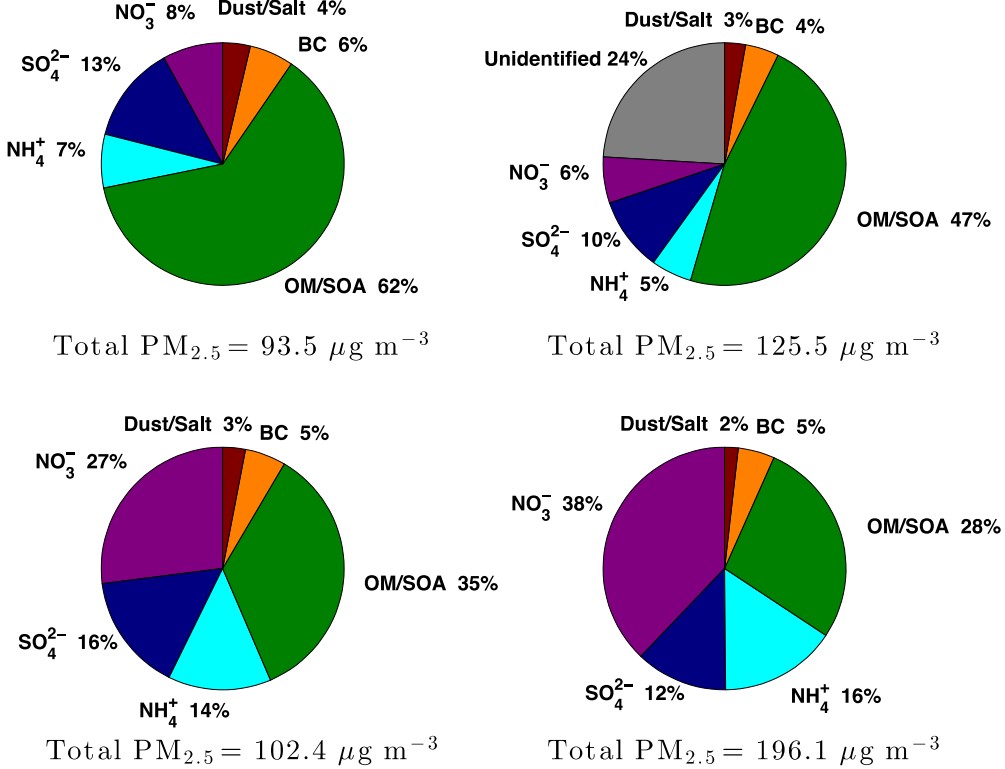

**Figure 3.** Fractional (%) mass contribution of each component (dust/salt, black carbon (BC), organic matter (OM), ammonium ($NH_4^+$), sulfate ($SO_4^{2-}$), and nitrate ($NO_3^-$)) to total $PM_{2.5}$ for the (a, b) observations (Ram and Sarin, 2011), where (a) excludes the "unidentified" component that is shown in (b), (c) AM4-CMIP5, and (d) AM4-CMIP6.

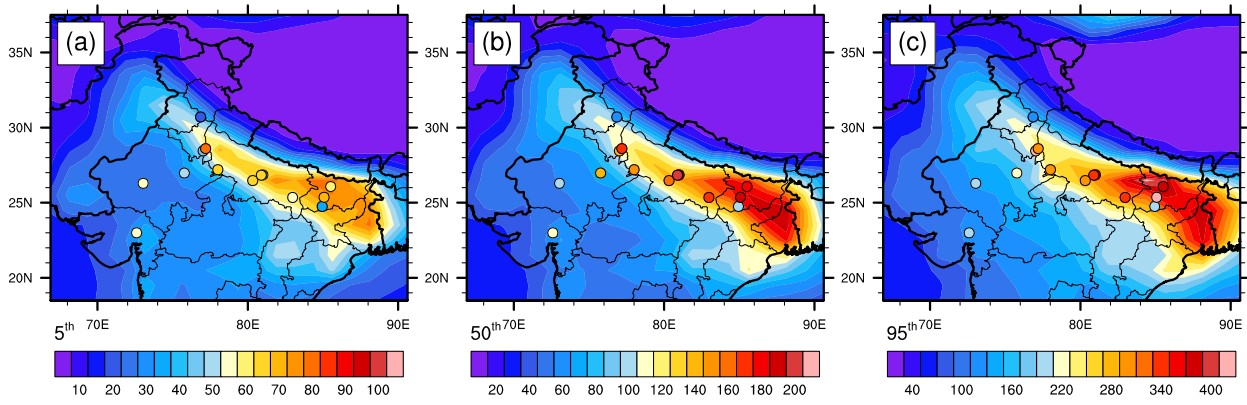

**Figure 4.** (a) 5th (b) 50th, and (c) 95th percentile of 24-h average $PM_{2.5}$ (µg m$^{-3}$) over 1 October 2015 – 31 March 2016 for the observations (circles), and the AM4-CMIP6-wet (background).



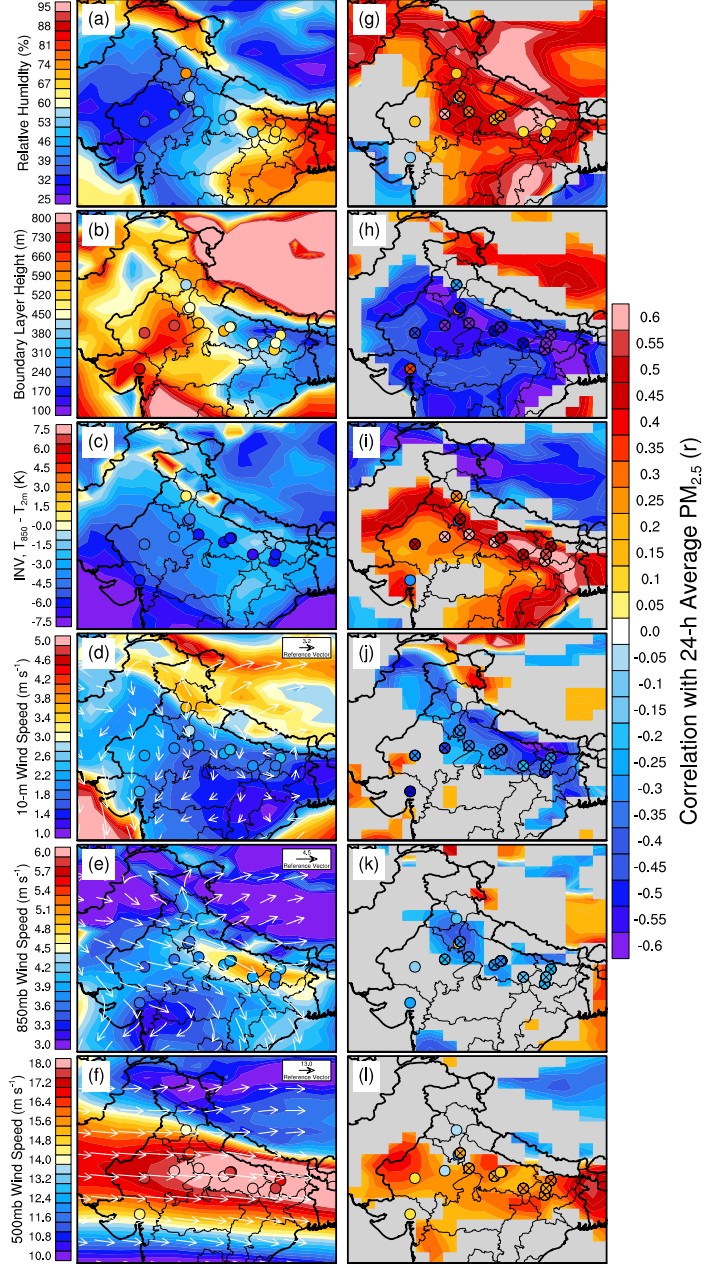

**Figure 5.** (a-f) Average meteorological conditions and (g-l) their correlations with daily average PM$_{2.5}$ for the observations (filled circles) and the AM4-CMIP6-wet (background) over 1 October 2015 – 31 March 2016. (a, g) relative humidity (RH, %), (b, h) boundary layer height (BLH, m), (c, i) temperature difference between 850 mb and 2-m (INV, K), (d, j) 10-m wind flow and speed (m s$^{-1}$), (e, k) 850 mb wind flow and speed (m s$^{-1}$), and (f, l) 500 mb wind flow and speed (m s$^{-1}$). For (g–l), only areas with correlations significant at the 95% confidence level based on a Student's T-test are shown for the model; circles with an 'x' denote the same for the observations.





**Figure 6.** (a-e) Monthly (October – March) and (f-j) diurnal cycles of (a, f) PM2.5, (b, g) relative humidity, (c, h) boundary layer height, (d, i) INV, and (e, j) 10 meter wind speed. The min-to-max range of the observations is shown in gray with the median in black. The median of grid cells containing the sites for the AM4-CMIP5 dry, AM4-CMIP6 dry, and AM4-CMIP6 wet are shown in blue, red, and green, respectively. Model-measurement correlations for the PM2.5 cycles are also shown.





**Figure 7.** Composite of anomalies of PM₂.₅ relative to monthly mean on days when ASI components (a) 10-m wind speed, (b) 500 mb wind speed, (c) precipitation, and (d) total ASI are met minus days when they are not during the period 1 October 2015 – 31 March 2016.





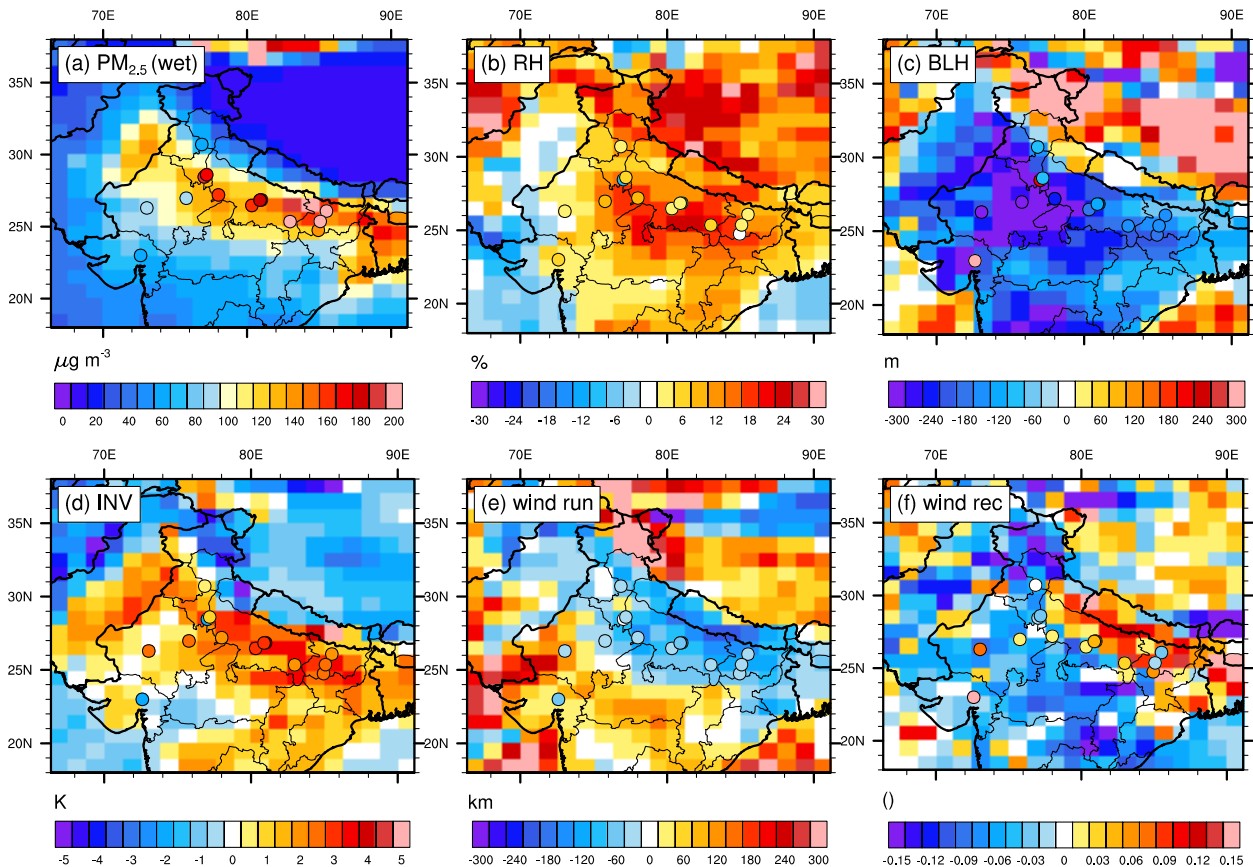

**Figure 8.** Composite of the 10 days with the highest $PM_{2.5}$ abundance minus the 10 days with the lowest from October 2015 – March 2016 for daily averages of (a) $PM_{2.5}$ (wet, µg m⁻³), (b) relative humidity (%), (c) boundary layer height (m), (d) temperature inversion strength (K), (e) wind run (km), and (f) wind recirculation (unitless) for the observations (circles) and the AM4-CMIP6-wet (background).