# Peer review of "Exploring the relationship between surface $PM_{2.5}$ and meteorology in Northern India"

_Atmospheric Chemistry and Physics, 2018_

## Referee Comment (RC1) · Anonymous Referee #1 · 28 Mar 2018

General comments

The authors applied the NOAA GFDL-AM4 model to simulate air quality over India for 20 years. They used a coarse horizontal resolution and two different emissions estimates (CMIP5 and CMIP6) in the model. They compared model predictions with observed PM25 in India for six winter months and performed detailed analysis. Model with CMIP5 substantially underestimated PM25 compared to observed data in Northern India. While the model with CMIP6 improved the predictions, it still underestimated PM25. Most monitoring stations in India are located in urban areas. The model with a coarse horizontal resolution is not suitable for examining PM25 in urban areas in India.

[Figure]

As such model under-prediction is expected. Finer scale resolution is needed. Several other issues need to be addressed.

Specific comments

Line 199-200 Reference is needed for the heterogeneous uptake coefficients used in the model.

Line 204-206 Several acronyms have already been defined before and are defined here and later. No need to define the acronyms multiple times.

Line 215-216 Dust1, dust2, ssalt1, ssalt2, ssalt3 are not defined in the article?

Line 261-266 and Figure 1 It is not clear if NO or NO2 emissions are shown in Figure 1. "NO" is used in one sentence but "NO2" is used in the other sentence. Need clarification. How NOx emissions are being speciated into NO and NO2 emissions?

Figure 2 Title of Figure 2 indicates "CMIP5-dry". However, legend shows "AM4-CMIP5 (wet)". Need clarification.

Figure 3 Observed data are taken from Kanpur site which is not clearly indicated in the Figure title.

Line 325-340 The model over-predicts aerosol nitrate substantially which may result from many factors including the use of high heterogeneous uptake coefficient for N2O5 (Table S1). Recent studies (Davis et al., 2008; Reimer et al., 2009; Brown et al., 2009; Phillips et al., 2016; Chang et al., 2016) suggest a much lower value for the heterogeneous uptake coefficient. A discussion of the impact of high heterogeneous uptake coefficient for N2O5 on model results is relevant.

Davis et al., 2008: Parameterization of N2O5 reaction probabilities on the surface of particles containing ammonium, sulfate, and nitrate, Atmos. Chem. Phys., 8, 5295–5311.

Riemer et al., 2009: The relative importance of organic coatings for the heterogeneous

hydrolysis of N2O5, JGR, 114.

Brown et al., 2009: Reactive uptake coefficients for N2O5 determined from aircraft measurements during the Second Texas Air Quality Study: Comparison to current model parameterizations, JGR, 114.

Phillips et al., 2016: Estimating N2O5 uptake coefficients using ambient measurements of NO3, N2O5, ClNO2 and particle-phase nitrate, ACP, 16, 13231-13249.

Chang et al., 2016: Evaluating N2O5 heterogeneous hydrolysis parameterizations for CalNex 2010, JGR: Atmosphere, 121, 5051–5070.

Line 443-444 The sentence suggests that annual cycle is shown in the Figure. However, it shows data for 6 months, not for the year.

Line 449-450 The sentence suggests model is biased high in RH. However, Figure 6b shows under-prediction of RH compared to observed data. Need clarification.

Line 492-497 Ram and Sarin (2011) analyzed measurement data and reported that nighttime aerosol nitrate level is five times greater than the day-time nitrate level. In contrast, this modeling study finds that aerosol nitrate peaks during mid-day. Despite the use of high uptake coefficient for N2O5, it finds that aerosol nitrate peaks during the day which reveals that HNO3 produced from the reaction of NO2 + OH likely dominates the production of aerosol nitrate. What caused the results to be completely opposite to that of Ram and Sarin (2011)?

Figure S7 Title is not clear. Figure d-f are missing.

Figure S8 Title refers to Figure 4 which should probably be Figure 5.

---

## Referee Comment (RC2) · Anonymous Referee #2 · 4 Apr 2018

The authors investigated the ability of an advanced new version of the NOAA GFDL-AM4 model to reproduce observed PM2.5 and its relationship to meteorology over Northern India. Considerable improvements in the meteorological part (modified topographic gravity wave drag parameterization, new double plume moist convection scheme, updated GFDL radiative transfer code) as well as in the chemistry and aerosol physics modules (updates in gas phase and heterogeneous chemistry, improved treatment of sulfate and nitrate chemistry, revised wet deposition scheme) were implemented in GFDL-AM4 and applied for the model simulations. Furthermore, increased horizontal and vertical resolutions compared to AM3 runs were used. A detailed analysis of the model results and comparison with measurements were performed to evalu-

ate the model performance and find out correlations of daily-averaged PM2.5 concentrations with meteorological variables. As a result, the spatial distribution of aerosol climatology and seasonal cycle simulated by the advanced AM4 has been significantly improved relative to AM3. The improved model reproduces most of the observed PM2.5–meteorology correlation patterns well, especially in the eastern IGP.

The authors provide a comprehensive and thorough study of the PM2.5 burden in Northern India due to the specific emissions as well as the extreme physical, chemical, and meteorological conditions in this area. The applied model NOAA GFDL-AM4 is state-of-the-art for the description of atmospheric chemistry-transport processes and regional climate modelling. The model results were analyzed and discussed extensively. However, due to the complexity of the modelling system and the relatively coarse grid resolution, such a discussion is difficult and sometimes speculative. It would therefore be desirable to verify several interesting findings by more detailed model runs with finer scale resolution for shorter selected periods. Altogether, the paper is sound, informative and well written. It can be published in the current form.

---

## Author Response (AR1)

The authors wish to thank each of the anonymous reviewers for taking the time to review our manuscript. While referee #2 found the manuscript to be publishable in its current, referee #1 has provided valuable input to the revised manuscript. We have responded (in blue) to each of the general and specific comments (in black) and hope they are to the satisfaction of the questions/concerned posed.

**Anonymous Referee #1**

General comments

The authors applied the NOAA GFDL-AM4 model to simulate air quality over India for 20 years. They used a coarse horizontal resolution and two different emissions estimates (CMIP5 and CMIP6) in the model. They compared model predictions with observed PM25 in India for six winter months and performed detailed analysis. Model with CMIP5 substantially underestimated PM25 compared to observed data in Northern India. While the model with CMIP6 improved the predictions, it still underestimated PM25. Most monitoring stations in India are located in urban areas. The model with a coarse horizontal resolution is not suitable for examining PM25 in urban areas in India.

We appreciate the reviewers concern with using this model to compare with urban air quality data in India. There is a long history of representation errors when testing gridded global model output against limited number of point observations of air quality – rural, urban, or otherwise. However, these errors exist even for relatively high resolution regional air quality models (e.g., 4–12 km) as sites are often clustered closely together. For example, there are 9 observing sites located in and around New Delhi and they are extremely diverse in their overall magnitude of $PM_{2.5}$ abundances (Figure 2a) despite being within ~25km of one another. In that sense, it's not clear how much more suitable a high resolution model would be. Furthermore, it would certainly prove difficult to obtain accurate emission data on the scale required, which is probably on the order of 1-5 km. Moreover, these are the only observations that are available and we attempt to make use of them as best as possible - not only to test the GFDL AM4 simulation of $PM_{2.5}$, but also compare modeled vs. observed meteorology-$PM_{2.5}$ relationships. In any case, we recognize this analysis has similar measurement-model mismatch issues that exist in previous works and we try to make this point in Section 3.1.

Specific comments

Line 199-200 Reference is needed for the heterogeneous uptake coefficients used in the model.

We have included a reference for the values in Table S1.

Line 204-206 Several acronyms have already been defined before and are defined here and later. No need to define the acronyms multiple times.

We have removed duplicate acronyms within the main text.

Line 215-216 Dust1, dust2, ssalt1, ssalt2, ssalt3 are not defined in the article?

Thank you for noting this. We have included a description of these components (the numbers correspond to different size bins; i.e., dust1-2 & ssalt1-3 are < 2.5µm)

Line 261-266 and Figure 1 It is not clear if NO or NO2 emissions are shown in Figure 1. "NO" is used in one sentence but "NO2" is used in the other sentence. Need clarification. How NOx emissions are being speciated into NO and NO2 emissions?

Thank you for catching this. NOx emissions are emitted as 100% NO and as such the references to $NO_2$ in the text and figure caption are incorrect, but they are merely a typo that we have corrected.

Figure 2 Title of Figure 2 indicates "CMIP5-dry". However, legend shows "AM4-CMIP5 (wet)". Need clarification.

Thank you for catching this. We have changed the caption for Figure 2 to read "AM4-CMIP5(wet)" as the legend indicates.

Figure 3 Observed data are taken from Kanpur site which is not clearly indicated in the Figure title.

We have added a description of the observational data (city name, lat, lon, elevation, time period, and reference) to the figure caption.

Line 325-340 The model over-predicts aerosol nitrate substantially which may result from many factors including the use of high heterogeneous uptake coefficient for N2O5 (Table S1). Recent studies (Davis et al., 2008; Reimer et al., 2009; Brown et al., 2009; Phillips et al., 2016; Chang et al., 2016) suggest a much lower value for the heterogeneous uptake coefficient. A discussion of the impact of high heterogeneous uptake coefficient for N2O5 on model results is relevant.

Davis et al., 2008: Parameterization of N2O5 reaction probabilities on the surface of particles containing ammonium, sulfate, and nitrate, Atmos. Chem. Phys., 8, 5295– 5311.

Riemer et al., 2009: The relative importance of organic coatings for the heterogeneous hydrolysis of N2O5, JGR, 114.

Brown et al., 2009: Reactive uptake coefficients for N2O5 determined from aircraft measurements during the Second Texas Air Quality Study: Comparison to current model parameterizations, JGR, 114.

Phillips et al., 2016: Estimating N2O5 uptake coefficients using ambient measurements of NO3, N2O5, ClNO2 and particle-phase nitrate, ACP, 16, 13231-13249.

Chang et al., 2016: Evaluating N2O5 heterogeneous hydrolysis parameterizations for CalNex 2010, JGR: Atmosphere, 121, 5051–5070.

Thank you for bringing this to our attention. The heterogeneous uptake coefficient used here for $N_2O_5$ is indeed significantly higher than those from recent studies, and has been the default value for the GFDL AM3 chemistry (Naik et al., 2013; Mao et al. 2013a). To test how our value affects the nitrate bias, as well as the potentially aberrant nitrate diurnal cycle (per the comment below), we have run an additional simulation that covers the 2008-2009 period of the observations by Ram et al. (2012) using the updated uptake value of 0.01, which is an order of magnitude smaller than our original. We have plotted the results of this simulation in Figure S10 (diurnal cycle of $PM_{2.5}$ components), which is shown below. Nitrate is shown in panel (f). Compare the blue line (base simulation of the Ram et al. period) with the red lines (sensitivity over the same period). The effect of the updated gamma value is to reduce nitrate abundances by ~15 ug m³ and ammonium abundances by about 5 ug m⁻³ with the largest changes at night. However, the diurnal cycle of nitrate (as with ammonium and sulfate) is qualitatively unchanged from the base simulation, with a relative maximum still occurring at midday. So, while we have reduced nitrate abundances, the seemingly aberrant diurnal cycle of nitrate is still evident even with the updated gamma and will require additional experimentation beyond the scope of this paper. We hope that this satisfies the reviewers comments/concerns. We have added the following to the paper to reflect the results of the sensitivity experiment:

*We test if the seemingly aberrant $NO_3^-$ diurnal cycle is a result of our choice of the value for the $N_2O_5$ heterogenous uptake coefficient (0.1), which is significantly higher than those reported by previous studies (e.g., Davis et al., 2008; Chang et al., 2016), by performing an additional simulation with an $N_2O_5$ uptake coefficient of 0.01 over period of the Ram et al. (2012) observations. The effect of the updated value is to reduce $NO_3^-$ by ~15 µg m⁻³, $NH_4^+$ by ~5 µg m⁻³, and $SO_4^{2-}$ by ~1 µg m⁻³, all with the largest changes at night (Fig S10d-f). However, the diurnal cycle of $NO_3^-$ (as with $NH_4^+$ and $SO_4^{2-}$) is qualitatively unchanged from the base simulation, with a relative maximum still occurring at midday. So, while we have reduced nitrate abundances, the midday $NO_3^-$ peak is still evident even with the updated gamma. One possible explanation is that the model prescribes monthly average deposition rates for $NH_4NO_3$ (i.e., no diurnal cycle), however, determining the cause of this midday peak will require additional experimentation beyond the scope of this paper.*

[Figure]

Line 443-444 The sentence suggests that annual cycle is shown in the Figure. However, it shows data for 6 months, not for the year.

Thank you, we have fixed the wording to denote that it is a monthly average for six months.

Line 449-450 The sentence suggests model is biased high in RH. However, Figure 6b shows under-prediction of RH compared to observed data. Need clarification.

Thank you for bringing this to our attention. We have fixed the sentence to reflect the low-bias.

Line 492-497 Ram and Sarin (2011) analyzed measurement data and reported that nighttime aerosol nitrate level is five times greater than the day-time nitrate level. In contrast, this modeling study finds that aerosol nitrate peaks during mid-day. Despite the use of high uptake coefficient for N2O5, it finds that aerosol nitrate peaks during the day which reveals that HNO3 produced from the reaction of NO2 + OH likely dominates the production of aerosol nitrate. What caused the results to be completely opposite to that of Ram and Sarin (2011)?

See discussion above.

Figure S7 Title is not clear. Figure d-f are missing.

We have fixed the figure caption. Only figures a-c are shown.

Figure S8 Title refers to Figure 4 which should probably be Figure 5.

Thank you for catching this, we have fixed the caption.

[revised manuscript text omitted]

Footer: Font: (Default) Times New Roman, 10 pt, Font color: Auto, English (UK), Justified, Widow/Orphan control, Border: Top: (No border), Bottom: (No border), Left: (No border), Right: (No border), Between : (No border), Tab stops: 3.13", Centered + 6.

| Page 1: [2] Style Definition | Schnell_et_al-ACP | 6/5/18 11:46:00 AM |
|---|---|---|

Balloon Text: Font: (Default) Tahoma, 8 pt, Font color: Auto, English (UK), Justified, Widow/Orphan control, Border: Top: (No border), Bottom: (No border), Left: (No border), Right: (No border), Between : (No border)

| Page 1: [3] Style Definition | Schnell_et_al-ACP | 6/5/18 11:46:00 AM |
|---|---|---|

Header: Font: (Default) Times New Roman, 10 pt, Font color: Auto, English (UK), Justified, Line spacing: 1.5 lines, Widow/Orphan control, Border: Top: (No border), Bottom: (No border), Left: (No border), Right: (No border), Between : (No border), Tab sto

| Page 1: [4] Style Definition | Schnell_et_al-ACP | 6/5/18 11:46:00 AM |
|---|---|---|

Heading 4: Font: (Default) Times New Roman, 10 pt, Font color: Auto, English (UK), Justified, Space Before: 0 pt, After: 0 pt, Add space between paragraphs of the same style, Line spacing: 1.5 lines, Widow/Orphan control, Don't keep lines together, Bor

| Page 1: [5] Style Definition | Schnell_et_al-ACP | 6/5/18 11:46:00 AM |
|---|---|---|

Heading 3: Font: (Default) Times New Roman, 10 pt, Font color: Auto, English (UK), Justified, Space Before: 12 pt, After: 12 pt, Add space between paragraphs of the same style, Widow/Orphan control, Don't keep lines together, Border: Top: (No border), B

| Page 1: [6] Style Definition | Schnell_et_al-ACP | 6/5/18 11:46:00 AM |
|---|---|---|

Heading 2: Font: (Default) Times New Roman, 10 pt, Font color: Auto, English (UK), Justified, Space Before: 12 pt, After: 12 pt, Add space between paragraphs of the same style, Widow/Orphan control, Don't keep lines together, Border: Top: (No border), B

| Page 1: [7] Formatted | Schnell_et_al-ACP | 6/5/18 11:46:00 AM |
|---|---|---|

Left: 0.65", Right: 0.65", Top: 0.39", Bottom: 0.93", Width: 8.27", Height: 9.45", Footer distance from edge: 0.51", From text: 0.16", Numbering: Restart each page

| Page 7: [8] Deleted | Schnell_et_al-ACP | 6/5/18 11:46:00 AM |
|---|---|---|

[Figure]

**Figure 1.** (**a-e**) Total CMIP6 anthropogenic emissions (Gg) of (**a**) black carbon (BC), (**b**) organic matter (OM), (**c**) $NO_2$, (**d**) $SO_2$, and (**e**) $NH_3$ over 1 October 2015 – 31 March 2016. Total emissions over the domain (in Tg) are

provided in the panel titles. **(f-j)** Difference (%) between CMIP6 and CMIP5 emissions (CMIP6 minus CMIP5) for the species in (**a-e**).

[Figure]

**Figure 2.** Time series of daily average PM$_{2.5}$ (1 October 2015 – 31 March 2016) for the grid cells over (**a**) New Delhi and (**b**) Kanpur/Lucknow, Uttar Pradesh. The min-to-max range of the multiple observations sites are shown in grey with their median in black. The number of sites is given in the panel titles. Modeled abundances are shown in (blue) CMIP5-dry, (red) CMIP6-dry, (green) CMIP6-wet, and (magenta) CMIP6-wet calculated with the GEOS-CHEM hygroscopic growth factors. The dashed vertical lines represent the 5-day festival of Diwali.

[Figure]

**Figure 3.** Fractional (%) mass contribution of each component (dust/salt, black carbon (BC), organic matter (OM), ammonium ($NH_4^+$), sulfate ($SO_4^{2-}$), and nitrate ($NO_3^-$)) to total PM2.5 for the (**a**, **b**) observations (Ram and Sarin, 2011), where (**a**) excludes the "unidentified" component that is shown in (**b**), (**c**) AM4-CMIP5, and (**d**) AM4-CMIP6.

[Figure]

**Figure 4.** (**a**) 5th (**b**) 50th, and (**c**) 95th percentile of 24-h average PM2.5 (µg m$^{-3}$) over 1 October 2015 – 31 March 2016 for the observations (circles), and the AM4-CMIP6-wet (background).

[Figure]

**Figure 5**. (**a-f**) Average meteorological conditions and (**g-l**) their correlations with daily average $PM_{2.5}$ for the observations (filled circles) and the AM4-CMIP6-wet (background) over 1 October 2015 – 31 March 2016. (**a**, **g**) relative humidity (RH, %), (**b**, **h**) boundary layer height (BLH, m), (**c**, **i**) temperature difference between 850 mb and 2-m (INV, K), (**d**, **j**) 10-m wind flow and speed (m s$^{-1}$), (**e**, **k**) 850 mb wind flow and speed (m s$^{-1}$), and (**f**, **l**) 500 mb wind flow and speed (m s$^{-1}$). For (**g–l**), only areas with correlations significant at the 95% confidence level based on a Student's T-test are shown for the model; circles with an 'x' denote the same for the observations.

[Figure]

**Figure 6.** (**a-e**) Monthly (October – March) and (**f-j**) diurnal cycles of (**a**, **f**) $PM_{2.5}$, (**b**, **g**) relative humidity, (**c**, **h**) boundary layer height, (**d**, **i**) INV, and (**e**, **j**) 10 meter wind speed. The min-to-max range of the observations is shown in gray with the median in black. The median of grid cells containing the sites for the AM4-CMIP5 dry, AM4-CMIP6 dry, and AM4-CMIP6 wet are shown in blue, red, and green, respectively. Model-measurement correlations for the $PM_{2.5}$ cycles are also shown.

[Figure]

**Figure 7.** Composite of anomalies of PM$_{2.5}$ relative to monthly mean on days when ASI components (**a**) 10-m wind speed, (**b**) 500 mb wind speed, (**c**) precipitation, and (**d**) total ASI are met minus days when they are not during the period 1 October 2015 – 31 March 2016.

[Figure]

**Figure 8.** Composite of the 10 days with the highest PM$_{2.5}$ abundance minus the 10 days with the lowest from October 2015 – March 2016 for daily averages of (**a**) PM$_{2.5}$ (wet, μg m$^{-3}$), (**b**) relative humidity (%), (**c**) boundary layer height (m), (**d**) temperature inversion strength (K), (**e**) wind run (km), and (**f**) wind recirculation (unitless) for the observations (circles) and the AM4-CMIP6-wet (background).